# Quasi-continuous transition from a Fermi liquid to a spin liquid in κ-(ET)$_2$Cu$_2$(CN)$_3$

Tetsuya Furukawa [1,2], Kazuhiko Kobashi[1], Yosuke Kurosaki[1], Kazuya Miyagawa[1] & Kazushi Kanoda [1]

The Mott metal-insulator transition—a manifestation of Coulomb interactions among electrons—is known as a discontinuous transition. Recent theoretical studies, however, suggest that the transition is continuous if the Mott insulator carries a spin liquid with a spinon Fermi surface. Here, we demonstrate the case of a quasi-continuous Mott transition from a Fermi liquid to a spin liquid in an organic triangular-lattice system κ-(ET)$_2$Cu$_2$(CN)$_3$. Transport experiments performed under fine pressure tuning have found that as the Mott transition is approached, the Fermi liquid coherence temperature continuously falls to the scale of kelvins, with a divergent quasi-particle decay rate on the metal side, and the charge gap continuously closes on the insulator side. A Clausius-Clapeyron analysis provides thermodynamic evidence for the extremely weak first-order nature of the transition. These results provide additional support for the existence of a spinon Fermi surface, which becomes an electron Fermi surface when charges are delocalized.

[1] Department of Applied Physics, University of Tokyo, 7-3-1 Hongo, Bunkyo-ku, Tokyo 113-8656, Japan. [2] Present address: Department of Applied Physics, Tokyo University of Science, Niijyuku 6-3-1, Katsushika-ku, Tokyo 125-8585, Japan. Correspondence and requests for materials should be addressed to T.F. (email: tetsuya.furukawa@rs.tus.ac.jp) or to K.K. (email: kanoda@ap.t.u-tokyo.ac.jp)

In a Mott insulator, mutually interacting spins are generally ordered at low temperatures. However, antiferromagnetic interactions are self-conflicting for spins on a triangle-based lattice, which may fall into exotic states called quantum spin liquids[1,2]. Indeed, spin liquids were substantiated in κ-$(ET)_2Cu_2(CN)_3$ and $EtMe_3Sb[Pd(dmit)_2]_2$ (ET and dmit denote BEDT-TTF=bis(ethylenedithio)tetrathiafulvalene and 1,3-dithiole-2-thione-4,5-dithiolate, respectively), which are organic materials with triangular lattices[3,4]; they exhibit unconventional spin excitations of a gapless or marginally gapped nature[3–9]. These properties lead to an intriguing proposal of delocalized spin excitations—spinons with a Fermi surface[10,11]—which has recently been discussed in a strong spin–orbit-coupled triangular-lattice $YbMgGaO_4$ system as well[12,13].

In both organic materials, the spin liquids reside near the Mott metal-insulator transition, which is accessible by applying pressure[14–17]. The vicinity to the Mott transition is theoretically suggested to be essential to the emergence of spin liquids in triangular lattices[2,10]. In a sense that any conventional symmetry breaking does not follow, the metal-insulator transition of spin liquids is a genuine Mott transition, at which only the charge degrees of freedom are frozen, as Mott originally conceptualized. The conventional Mott transition is a first-order transition with a clear discontinuity, as shown by various experiments[18] and the celebrated dynamical mean-field theory (DMFT)[19]. However, if the spin excitations in the spin liquid are delocalized as suggested theoretically, the itineracy of the electrons is lost in the charge sector on the Mott transition but persists in the spin sector as fractionalized spinons that hold a Fermi surface, leading to an unconventional Mott localization of electrons; indeed, a continuous Mott transition distinct from the conventional cases is predicted[20–22]. Thus, a close examination of the Mott transition is expected to provide insight into the fractionalization of particles in condensed matter.

Here, we investigate the charge transport of κ-$(ET)_2Cu_2(CN)_3$ under precisely controlled pressures to reveal how the Fermi liquid and the spin liquid meet in the pressure (P)–temperature (T) plane. We present that the Fermi liquid coherence temperature continuously falls to the scale of kelvins, with a divergent quasi-particle decay rate on the metal side, and the charge gap gradually closes on the insulator side. In addition, a Clausius-Clapeyron analysis of the P–T phase diagram yields thermodynamic evidence for the extremely weak first-order nature of the transition. These findings provide additional support for the existence of a spinon Fermi surface in the spin liquid phase.

## Results

**Pressure–temperature phase diagram.** κ-$(ET)_2Cu_2(CN)_3$ is a layered organic system composed of conducting ET layers and insulating $Cu_2(CN)_3$ layers. In the conducting layers, the ET molecules form dimers, which constitute a nearly isotropic triangular lattice with a half-filled band[23] (Supplementary Fig. 3). The in-plane resistance was measured with the standard dc four-probe method. Hydrostatic pressure was applied by using, as pressure media, helium gas for $P < 200$ MPa and Daphne 7373 oil or DEMNUM S-20 oil for $P > 200$ MPa. Experimental details are described in Methods.

The main points of the present results for κ-$(ET)_2Cu_2(CN)_3$ are featured in the P–T phase diagram (Fig. 1a–c), which was constructed on the basis of the resistivity data described below (Fig. 2a–d). The first-order transition curve separating the spin liquid insulator and a metal (or a superconductor) shows a characteristic bending profile (Fig. 1a) and leads to a fan-shaped quantum critical region at high temperatures[15] (Fig. 1b, c). The temperature region of a Fermi liquid, in which the resistivity follows the $T^2$ law, is strongly suppressed as the Mott boundary is approached. The proximity of the superconducting phase to the spin liquid phase is confirmed. Figure 2a shows the temperature dependence of resistance for $P = 0$–202 MPa, where a systematic variation from an insulating behaviour to a metallic behaviour emerges. For $P \geq 135$ MPa, the resistance exhibited jumps with barely recognizable hysteresis, indicating a weak first-order Mott

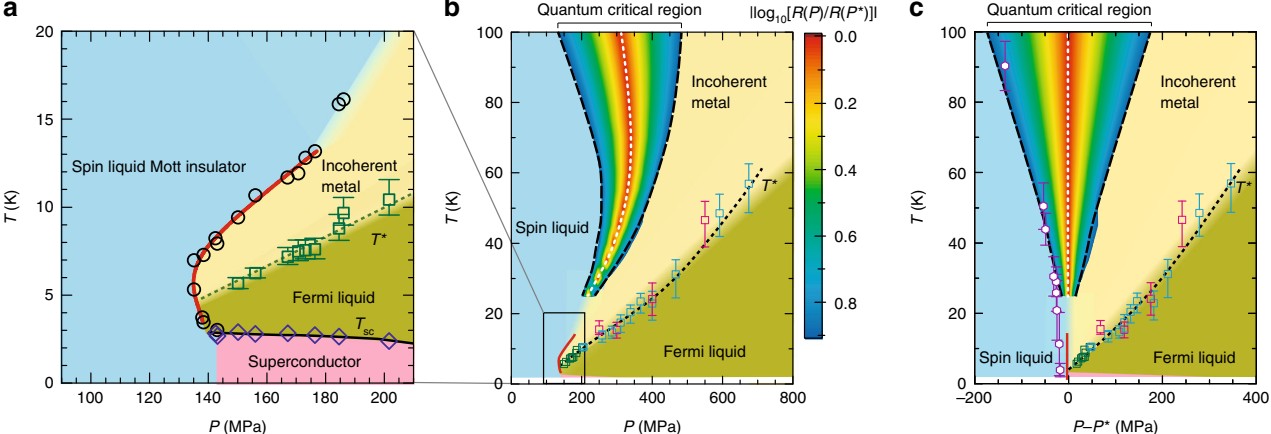

**Fig. 1** Pressure–temperature phase diagram of κ-$(ET)_2Cu_2(CN)_3$. **a** Low-temperature phase diagram in the vicinity of the Mott transition constructed from the resistance behaviours of samples #1 and #2. The black open circles indicate the points at which resistive jumps associated with the first-order Mott transition were observed, and the red bold line is a fit to the data points and was used in the entropy analyses (Fig. 4a–d). The blue diamonds indicate the onset of the superconducting transition $T_{sc}$. The green squares indicate the coherence temperatures $T^*$. We defined $T^*$ as the temperature at which the deviation in resistance $R(T)$ from the $T^2$ law characteristic of the Fermi liquids, $R_{fit}(T) = R_0 + AT^2$, exceeded 10% of $AT^2$, which fits the low-temperature data (see Fig. 2b and Supplementary Fig. 4). We evaluated the upper and lower bounds of errors in $T^*$ as the temperatures yielding 15 and 5% deviations of $R(T)$, respectively. **b** Phase diagram over a wide temperature–pressure range. The squares indicate the $T^*$ values determined from the resistance behaviours of samples #1 (green), #3 (pink) and #4 (blue). The white broken line indicates the Mott crossover pressure (the Widom line), $P^*(T)$, taken from ref. [15]. The fan-shaped contour plot, which is reproduced from ref. [15], represents the magnitude of $|\log_{10} R(P)/R(P^*)|$ in the quantum critical region. **c** Phase diagram in the $P$–$P^*$–$T$ plane. The open hexagons indicate the charge gaps (see Fig. 3) divided by a factor of three, $\Delta/3$, in order to see the correspondence with the pressure profile of the quantum critical energy scale. All of the energy scales characteristic of the metallic and insulating phases moved to collapse towards the Mott transition point, as seen in the figure

transition; we define $P_c = 135$ MPa as the critical pressure of the Mott transition at low temperatures. The non-monotonic temperature dependence of resistance near 120 MPa, which appears strange, is reasonable, considering the bending of the quasi-continuous Mott transition line towards higher pressures with increasing temperature. As the temperature decreases from approximately 15 K to several kelvins at a fixed pressure of $P \lesssim 135$ MPa, the system approaches the metallic phase because the bent first-order transition line ($dT/dP > 0$) nears the fixed pressure at low temperatures (see Fig. 1a). In addition, the present Mott transition has a quasi-continuous nature. Thus, the approach to the Mott boundary by cooling causes a decrease in the charge gap in the insulating side and possibly results in a decrease in resistance with temperature. Eventually, the resistance increases on further cooling because the system is located in the Mott insulating side at a finite distance from the Mott transition line, which is nearly vertical at temperatures below ~5 K. An absence of a resistance jump for $T \gtrsim 20$ K (Fig. 2d) means that the critical endpoint of the Mott transition is located between 16 and 20 K, which is approximately half of the critical endpoint in κ-(ET)$_2$Cu [N(CN)$_2$]Cl, 40 K[24].

**Critical behaviour on the metal side.** Figure 2b shows the resistance in the metallic phase plotted against $T^2$. The $T^2$ dependence expected in a conventional Fermi liquid, $R(T) = R_0 + AT^2$, holds over a limited low-temperature range below the coherence temperature $T^*$, which is pushed down to the order of kelvin near the Mott boundary (Fig. 1a–c). The three orders of magnitude drop of $T^*$ from the bare energy scale of the bandwidth, ~0.5 eV (~6000 K)[16], suggests that the system is situated

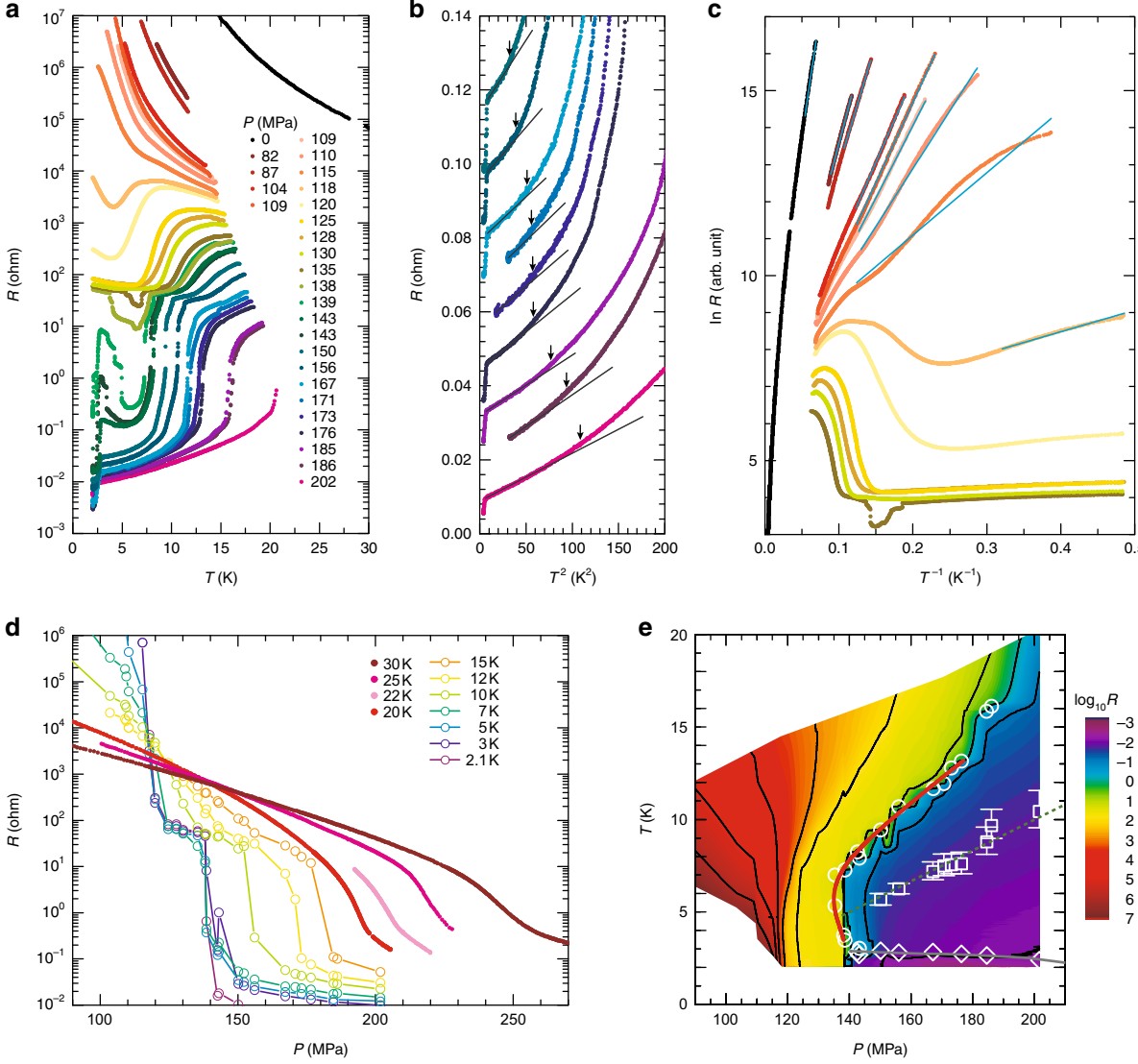

**Fig. 2** Transport properties of κ-(ET)$_2$Cu$_2$(CN)$_3$ near the Mott transition. **a** Temperature dependence of the resistance of sample #1. The presented data are for low temperatures below the melting point of the helium pressure medium because the solidification is followed by a pressure drop, causing a spurious change in resistivity (see Supplementary Note 1 and Supplementary Fig. 2). **b** Resistance (offset by an interval of 0.01 ohm) vs. $T^2$. The solid lines are fits of form $R(T) = R_0 + AT^2$ to the data. The coherence temperature $T^*$ at each pressure is indicated by arrows (see Fig. 1 for the definition). **c** Arrhenius plot of resistance in the insulating phase. The cyan solid lines are the fits of form $\ln R(T) = \Delta/2 \times T^{-1} + \text{const.}$ to the data at low temperatures (see Supplementary Note 6 and Supplementary Fig. 9 for details). **d** Pressure dependencies of the resistances of samples #1 ($T \leq 22$ K) and #2 ($T \geq 25$ K) at fixed temperatures. The resistance of sample #2 was multiplied by 1.5 for comparison. The measurements at $T$ above 20 K, where the helium pressure medium is in a liquid state, were performed in isothermal pressure sweeps. **e** Contour plot of the logarithmic resistance of sample #1 in the $P$–$T$ plane. The black lines are contour lines. The open circles (Mott transition points), squares ($T^*$) and diamonds ($T_{sc}$) are reproduced from Fig. 1a

near the quantum critical regime. Referring to theoretical investigations[21,22], $T^*$ might correspond to the crossover temperature between a Fermi liquid and marginal Fermi liquid, in which the quasi particles are scattered by critical fluctuations of gauge field. The locations of $T^*$ and the quantum critical region in the pressure–temperature phase diagram (Fig. 1c) are overall consistent with the theoretical prediction. The coefficient $A$ of the $T^2$ term, normalized to the room temperature resistance value of each sample, is enhanced as the Mott boundary is approached (Fig. 3), indicating an increase in quasi-particle decay rate. The divergent behaviour of $A$ persists in the very vicinity of $P_c$, $|P-P_c|$ < 50 MPa, whereas the increase in $A$ becomes weak in the vicinity of the conventional Mott transition (see Supplementary Fig. 8). The pressure dependence of $A$ is well fitted by a power-law, $A \propto |P-P_c|^{-0.75 \pm 0.01}$, which does not show an excellent agreement with the theoretical prediction[21] of the quasi-particle decay rate near the quantum Mott transition, $\gamma_{qp} \propto |P-P_c|^{-2\nu}$ ($2\nu \sim 1.34$). The gentler divergence in the experiment than in the theory may reflect the weak first-order nature of the transition. In conventional correlated electron systems, $A$ is proportional to the square of effective mass, and thus a power-law divergence of $A$ is linked to a power-law divergence of the effective mass. Note, however, that this may not be the case in the quantum Mott transition of a spin liquid, according to quantum-field[21] and phenomenological[25] theories; namely, the effective mass may not diverge in a power-law manner.

**Critical behaviour on the insulator side.** In the insulating phase, the resistance is characterized by a charge gap, $\Delta$, that is deduced from the line fitting to the data of $2d(\ln R)/d(1/T)$ at low temperatures (see Supplementary Note 6 and Supplementary Fig. 9 for details). As shown in Figs. 2c, 3, a remarkable feature is the gradual closing of the charge gap following $\Delta \propto |P-P_c|^{0.73 \pm 0.04}$ for $|P-P_c| \gtrsim 25$ MPa, which suggests the quasi-continuous nature of the Mott transition. This form is close to the theoretical prediction, $\Delta \propto |P-P_c|^{0.67}$, for the quantum Mott transition[21,22], whereas some numerical studies suggest $\Delta \propto |P-P_c|$[26,27,]. The anomalous drop in $\Delta$ for $|P-P_c|$ < 25 MPa implies marginally gapped or gapless charge excitations in the vicinity of the Mott transition;

indeed, the resistance shows a plateau in the narrow range of pressures around the Mott boundary below 10 K (Fig. 2a, d). To visualize these resistance behaviours, the logarithm of resistance is indicated by a range of colours on the $P-T$ phase diagram (Fig. 2e), which shows that the (yellow-coloured) plateau region resides in the lower-pressure side of the Mott boundary. Remarkably, such a resistance plateau is theoretically suggested to appear at the two-dimensional quantum Mott transition of a spin liquid[21,22]. The predicted sheet resistance at the plateau is ~ 8 $h/e^2$ ($h$ is the Plank constant, and $e$ is the elementary charge), which is smaller than the experimental sheet resistance of one conducting layer in the plateau, ~33 $h/e^2$ (Supplementary Note 4). In the experiments, the plateau appeared in a narrow but finite pressure region, whereas it occurs only at a critical pressure and is fan shaped at high temperatures in the theoretical prediction. The disorder in real systems can extend the critical point into a finite region, which may also explain the discrepancy in the magnitude of the sheet resistance. It is suggested that disorder in the polymeric atomic configurations in the anion layers affects the charge transport at ambient pressure[28].

**Clausius-Clapeyron analysis of the $P-T$ phase diagram.** As seen above, the transport characteristics around the Mott boundary are featured by a substantial falloff in $T^*$, critical behaviours of the coefficient $A$, closing of charge gap $\Delta$, and resistance plateau, all of which go against the conventional first-order Mott transition but accord with the theoretical consequences of the Mott transition of a spin liquid with a spinon Fermi surface[20–22]. Below, we discuss the thermodynamics of the Mott phase diagram in terms of the Clausius-Clapeyron relation, which relates the slope of the phase boundary, $\partial P_{jump}/\partial T_{jump}$, to the entropy difference between the neighbouring phases through $S_{QSL}-S_{metal} = \Delta V \times \partial P_{jump}/\partial T_{jump}$, where $S_{QSL}$ and $S_{metal}$ are the molar entropy of the quantum spin liquid (QSL) and the metal, respectively, and $\Delta V$ ($=V_{QSL}-V_{metal}$) is the molar volume change. Assuming that the lattice contributions to $S_{QSL}$ and $S_{metal}$ do not largely differ, $S_{QSL}-S_{metal}$ is approximately equal to $S_{QSL,spin}-S_{metal,el}$, where $S_{QSL,spin}$ ($S_{metal,el}$) is the spin (electron) contribution to $S_{QSL}$ ($S_{metal}$); the validity of this assumption is argued later. The $\partial P_{jump}/\partial T_{jump}$ determined from the fitting curve of the phase boundary (Fig. 1a) gives the temperature dependence of $S_{QSL,spin}-S_{metal,el}$ with $\Delta V$ as a parameter (Fig. 4a). We also performed a Clausius-Clapeyron analysis while incorporating the temperature dependence of $\Delta V$ and found the results in Fig. 4 to be practically unaltered (see Supplementary Note 5 and Supplementary Fig. 6 for details). Because $\Delta V$ and $P$ are thermodynamically conjugate variables, $\Delta V$ plays a role as an order parameter of Mott transition and therefore measures the strength of the first-order transition.

For evaluating $S_{QSL,spin}$, $S_{metal,el}$ is reasonably assumed to be $\gamma T$, where $\gamma$ is estimated to be ~ 27.6 mJ K$^{-2}$ mol$^{-1}$, referring to the spin susceptibility of $\chi_{metal} = 3.8 \times 10^{-4}$ emu mol$^{-1}$ (ref. 29) and Wilson ratio of $R_W = 1$, which holds in the metallic phases of κ-type compounds (Supplementary Note 2). The entropy of the spin liquid is obtained through $S_{QSL,spin} = \gamma T + \Delta V \times (\partial P_{jump}/\partial T_{jump})$, as displayed for several values of $\Delta V$ in Fig. 4b. $S_{QSL,spin}$ decreased on cooling, particularly below 10 K, and fell below $S_{metal,el}$ at approximately 6 K (Fig. 4b). The rather steep entropy loss at approximately 6 K implies that the 6 K anomaly observed at ambient pressure[5,6,30] persists up to the Mott boundary. Because $S_{QSL,spin}$ must not increase on cooling, $\Delta V$ should be ~2 × 10$^{-8}$ m$^3$ mol$^{-1}$ or less (Fig. 4b). We also estimated the specific heat from $S_{QSL,spin}$, as shown in Supplementary Fig. 7. Although the specific heat is less reliable since it is given by the second derivative of the fitting curve of the phase boundary, the

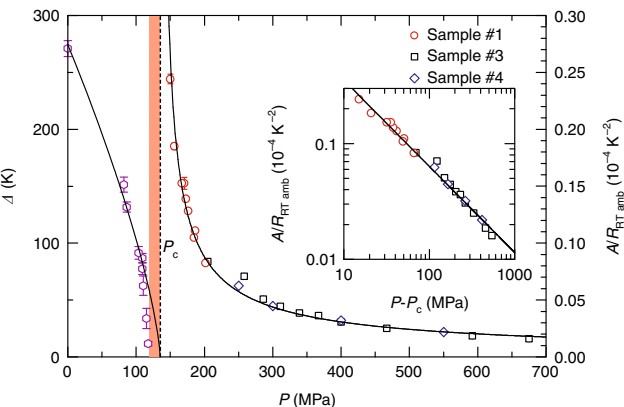

**Fig. 3** Pressure dependence of the charge gap and the $T^2$ coefficient of resistance. The charge gap $\Delta$ was deduced from the line fitting to the data of $2d(\ln R)/d(1/T)$ at low temperatures, and the error of the fitting constant is also indicated (see Supplementary Note 6 and Supplementary Fig. 9 for details). The solid lines indicate the fitting curves of $\Delta \propto |P-P_c|^{0.73 \pm 0.04}$ and $A \propto |P-P_c|^{-0.75 \pm 0.01}$, where $P_c$ is 135 MPa. The $T^2$ coefficient $A$ is normalized by resistance at room temperature under ambient pressure, $R_{RT,amb}$, for each sample. The charge gap is almost closed in the orange-coloured pressure region. The inset shows a log–log plot of $A/R_{RT,amb}$ vs. $P-P_c$

deduced specific heat appears to show a qualitative feature, that is, peak formations at approximately 6–7 K, akin to the ambient-pressure behaviour[6] and suggesting that the 6 K anomaly at ambient pressure persists up to the critical pressure of the Mott transition.

For comparison, we performed a similar analysis for κ-(ET)$_2$Cu[N(CN)$_2$]Cl, which exhibits an antiferromagnetic order in the Mott insulating phase (Fig. 4c, d and Supplementary Fig. 5). The spin entropy of the antiferromagnetic insulator (AFI) side, $S_{AFI, spin}$, rapidly decreases with temperature, even falling below $S_{metal,el}$ at as high as 30 K, irrespective of the choice of $\Delta V$ value. In the insulating side on the Mott boundary, spins undergo an antiferromagnetic order at approximately $T_N = 15\,K$[31]. Thus, spins lose their entropy from much higher temperatures than the Néel order due to short-range ordering, as shown by the thermodynamic and magnetic measurements at ambient pressure[32,33]. The requirement that the entropy should die out rapidly at low temperatures narrows down the possible values of $\Delta V$ to a range of $4$–$6 \times 10^{-7}\,m^3\,mol^{-1}$ (Fig. 4d), which is in good agreement with the experimentally determined value of ~$5.4 \times 10^{-7}\,m^3\,mol^{-1}$ (ref. [34], also see Supplementary Note 3) and thus corroborates the validity of the present analysis. Remarkably, the $\Delta V$ of κ-(ET)$_2$Cu$_2$(CN)$_3$ is several tens of times smaller than the $\Delta V$ of κ-(ET)$_2$Cu[N(CN)$_2$]Cl, which provides thermodynamic evidence for the extremely weak first-order nature of the Mott transition in κ-(ET)$_2$Cu$_2$(CN)$_3$, in accordance with the quasi-continuous nature of the transition revealed by the charge transport. Note that the large value of $\Delta V$ in κ-(ET)$_2$Cu[N(CN)$_2$]Cl holds even above $T_N$, meaning that the extremely small $\Delta V$ value in κ-(ET)$_2$Cu$_2$(CN)$_3$ is not attributable only to the paramagnetic nature of the Mott insulator.

It is suggestive that the temperature of the inflection point in the entropy decrease, 6 K, roughly coincides with $T^*$ in the metallic side at the Mott boundary. A similar anomaly suggesting the rapid release of the spin entropy was observed in the specific heat at approximately 6 K at ambient pressure[6], and the formation of a spinon Fermi surface from an incoherent paramagnetic state was argued as a possible origin. The present results may suggest that the electron Fermi surface in the metallic phase and the spinon Fermi surface in the spin liquid phase take shape at similar temperatures, although the origin of the 6 K anomaly at ambient pressure is under debate.

## Discussion

The charge transport in κ-(ET)$_2$Cu$_2$(CN)$_3$ and the thermodynamic arguments consistently highlight the quasi-continuous nature of the Mott transition, which contradicts the conventional picture but is reconciled with the theoretical consequences of a transition from an electron Fermi surface to a spinon Fermi surface. We note, however, that the correspondence between the experiments and theories based on the Hubbard model is not exact in that κ-(ET)$_2$Cu$_2$(CN)$_3$ exhibits a first-order transition, although it is extremely weak. This may indicate that some additional factors not included in the theoretical treatments affect the ground states; indeed, superconductivity appears in the metallic phase, and the 6 K anomaly is possibly suggestive of some instability of the spin liquid emerging in the insulating phase. In addition, the possible first-order transition due to long-range interactions among electrons is theoretically discussed[21].

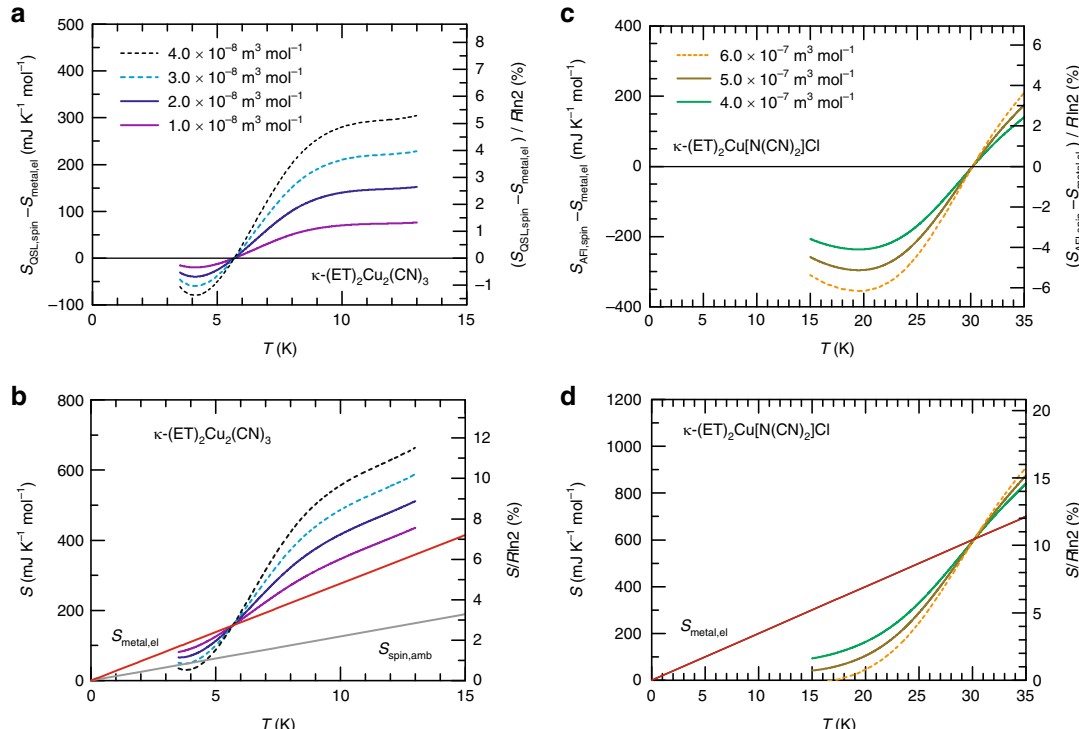

**Fig. 4** Thermodynamic properties of the spin liquid on the Mott boundary. The temperature dependencies of **a** $S_{QSL,spin} - S_{metal,el}$ and **b** $S_{QSL,spin}$ for κ-(ET)$_2$Cu$_2$(CN)$_3$ and **c** $S_{AFI,spin} - S_{metal,el}$ and **d** $S_{AFI,spin}$ for κ-(ET)$_2$Cu[N(CN)$_2$]Cl are indicated for several values of the parameter $\Delta V$. $R$ is the universal gas constant. The dashed lines are unphysical results that do not fulfil the fundamental relations $S > 0$ and $dS/dT > 0$ owing to too large $\Delta V$ values, which give the upper bounds for realistic $\Delta V$ values. The grey line in **b** indicates the linearly temperature-dependent spin entropy at ambient pressure $S_{spin,amb} = \gamma_s T$, with $\gamma_s = 12.6\,mJ\,K^{-2}\,mol^{-1}$ (ref. [6]). The red lines in **b** and **d** show $S_{metal,el} = \gamma T$, where $\gamma = 27.6\,mJ\,K^{-2}\,mol^{-1}$ for κ-(ET)$_2$Cu$_2$(CN)$_3$, and $\gamma = 20\,mJ\,K^{-2}\,mol^{-1}$ for κ-(ET)$_2$Cu[N(CN)$_2$]Cl. The $\gamma$ value in the latter is that of the partially deuterated κ-(ET)$_2$Cu[N(CN)$_2$]Br[36], which is a Mott insulator on the verge of the Mott transition

Very recently, a DMFT-based theoretical treatment incorporating spinon excitations suggests the first-order nature of the Mott transition of a spinon Fermi surface[35]. The faint remanence of the first-order nature may have some relevance to these theoretical consequences[21,35].

Finally, we discuss the relation between the low-temperature (low-$T$) quantum criticality observed here and the high-temperature (high-$T$) quantum criticality (Fig. 1b, c). The latter has the following key features. First, it is commonly observed for various systems irrespective of lattice geometry, whether the Mott insulator is a spin liquid or an antiferromagnet or whether the Mott transition is of the first order or quasi-continuous. Second, the high-$T$ regime should have no memories about Fermi surfaces in the metallic side and the nature of the magnetic ground state in the Mott insulating side. Third, the high-$T$ quantum critical scaling of resistivity is reproduced by the single-site DMFT of the Hubbard model. All the above suggest that the high-$T$ quantum criticality is a universal high-energy characteristic of the Mott transition. In contrast, the low-$T$ quantum criticality is likely specific to the fluctuations between electron Fermi surfaces and spinon Fermi surfaces, being closely related to the kinds of ground states. In our tentative view, the Mott transition has a hierarchal energy structure. The high-energy quantum criticality is generic, whereas the low-energy one depends on the details of the lattice, and the former is characterized by the scaling function fulfilled by resistivity in contrast to the constant resistivity in the latter. The proximity of the $z\nu$ values of the two criticalities may provide some insight into their possible connection, which awaits theoretical exploration.

## Methods

**Resistance measurements under pressure**. Single crystals of $\kappa$-(ET)$_2$Cu$_2$(CN)$_3$ (named #1, #2, #3 and #4) were grown by the conventional electrochemical method. The in-plane resistivity was measured with the d.c. four-probe method. Gold wires with diameters of 15–25 μm were glued onto the crystal faces with carbon paste to serve as electrodes. To apply hydrostatic pressure, we used pressure media of helium (for samples #1 and #2) for $P < 200$ MPa, Daphne 7373 oil (for sample #3) and DEMNUM S-20 oil (for sample #4) for $P > 200$ MPa. These pressure media are solidified at low temperatures. The melting temperature of helium is as low as 0–20 K for the pressures studied here (Supplementary Fig. 1), whereas those of the two oils exceed 100 K. Upon solidification of the pressure media, the pressures in the cell are somewhat reduced. The pressure reduction for the helium medium, which is in a range of 10–30 MPa, could be precisely evaluated using the method described in Supplementary Note 1. Sample #1 was studied across the helium solidification curve in Supplementary Fig. 1; the pressure was determined by the method with an accuracy of 1 MPa. Sample #2 was measured at higher temperatures than the helium solidification curve. For samples #3 and #4 in the oil media, the pressure values quoted in this article are those obtained by subtracting 300 MPa (sample #3) or 250 MPa (sample #4) from the pressures monitored at room temperature.

**Data availability**. The data that support the plots within this paper and other findings of this study are available from the corresponding author upon reasonable request.

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

## Acknowledgements

This work was supported in part by JSPS KAKENHI under Grant Nos. 20110002, 25220709, 24654101 and 17K05532 and the US National Science Foundation under Grant No. PHYS-1066293 and the hospitality of the Aspen Center for Physics.

## Author contributions

T.F. and K. Kanoda designed the experiments. T.F., K. Kobashi and Y.K. performed the resistance measurements. K.M. grew the single crystal for the study. All authors interpreted the data. T.F. wrote the manuscript with the assistance of K.M. and K. Kanoda. K. Kanoda headed this project.

## Additional information

**Competing interests:** The authors declare no competing financial interests.

