## [Peer Review File · Nature Communications]

Reviewers' comments:

Reviewer #1 (Remarks to the Author):

The paper entitled "Quasi-continuous transition from a Fermi liquid to a spin liquid" proposes an analysis of a Mott transition, both pressure and temperature dependent, in an organic material with triangular lattice.

The Authors interpret some of their transport results as typical signatures of the Mott transition from a Fermi liquid to a spin liquid with a spinon Fermi surface.

Even if the subject is of current interest, important revisions as discussed thereafter are at least required in order to clarify the demonstration and to make it more convincing.

One can read at l. 126: "...the transport characteristics around the Mott boundary are featured by a substantial falloff in T^* , the critical behaviors of the coefficient A , the closing of charge gap Δ , and the resistance plateau, all of which go against the conventional first-order Mott transition but accord with the theoretical consequences of the Mott transition of a spin liquid with a spinon Fermi surface."

The decrease of T^* and the divergent behavior of the coefficient A are typical signatures for conventional first-order Mott transition both related to the divergency of the electronic effective mass as predicted theoretically (Rev. Mod. Phys. 68, 13–125 (1996) and references therein) and demonstrated in several experiments.

As a consequence, only "the closing of charge gap Δ and the resistance plateau go against the conventional first-order Mott transition but accord with the theoretical consequences of the Mott transition of a spin liquid with a spinon Fermi surface".

The related text in the article needs to be changed accordingly.

It results then that the closing of charge gap Δ and the resistance plateau are two crucial features calling for improved analysis and discussions in order to demonstrate a "Quasi-continuous transition from a Fermi liquid to a spin liquid".

Concerning the closing of charge gap, Fig. 2c displays an Arrhenius plot of resistance in the insulating phase with $\ln(R)$ as a function of $1/T$. One cannot judge the validity of the performed fits especially in the regions of interest, namely where the charge gap is small. A plot of the derivative of resistance as $2 \cdot d(\ln(R))/d(1/T)$ as a function of temperature (namely the gap as a function of T), or equivalent representation, could better demonstrate the vanishing of the gap.

In addition, Fig. 3 shows the pressure dependence of the gap with very small values in the regions around 120 MPa, including likely $\Delta/2 < T$. In such a regime, resistance is usually not activated because thermal excitations are higher than the charge gap. Thus, the authors have to explain the method used to extract the gap within this regime of small values in order to demonstrate its vanishing.

Concerning the resistance plateau, the reported value is higher than the predicted one as stated by the Authors but this resistance behavior appears qualitatively distinct from the resistance behavior across a conventional Mott transition. Then, it could be consistent with a Mott transition from a Fermi liquid to a spin liquid at a qualitative level. Nevertheless, a non monotonic resistance behavior is clearly seen in Fig. 2a both below and above 120 MPa (with a relative maximum of $R(T)$) which is neither really discussed by the Authors and nor theoretically expected.

In particular, there is a sizable decrease of resistance in the Mott insulating state at 118 MPa for instance as a function of decreasing temperature which is very unusual and requires at least some comments from the Authors.

Furthermore, the Authors reproduce in Fig. 1b and 1c a Quantum critical region over a wide

temperature range. The Authors should precise how this high temperature Quantum critical region is connected to the low temperature Quantum critical region as theoretically discussed in Phys. Rev. B 86, 321 245102 (2012), the latter being defined with a temperature independent resistance not supported by experimental results over such a wide temperature range. The suggested role of disorder at l. 122 to explain all the discrepancies with theoretical predictions needs to be clarified in order to provide a consistent description of the phase diagrams.

Finally, the Authors propose an analysis of thermodynamic properties of spin liquid on the Mott boundary by extracting entropy, based on several assumptions and considering the volume change as a constant free parameter.

Also, one can read at l. 138: "The $dP_{\text{jump}}/dT_{\text{jump}}$ determined from the fitting curve of the phase boundary (Fig. 1a) gives the temperature dependence of $S_{\text{QL,spin}} - S_{\text{metal,el}}$ with ΔV as a parameter (Fig. 4a). Because ΔV and P are thermodynamically conjugate variables, ΔV plays a role of order parameter of Mott transition and therefore measures the strength of the first-order transition."

If the volume change "plays a role of order parameter of Mott transition" as stated, it must be temperature dependent in contrast to the assumption made by the Authors which has allowed them to infer the temperature dependence of entropy in Fig. 4. Therefore, the Authors have to reconcile their point of view with their analysis or at least precise that the analysis of the deduced entropy must only be considered at a qualitative level despite its interest.

Reviewer #2 (Remarks to the Author):

The article by T. Furukawa *et al.* reports electric transport property of organic charge transfer complex, κ -(BEDT-TTF)₂Cu₂(CN)₃ under precisely tuned external pressures in order to study the critical phenomenon around the Mott boundary between spin liquid phase and metallic (superconductive) phase. From the thermodynamic analysis of the p - T phase diagram determined by the conductivity as a function of T with varying p , they concluded that the Mott boundary has weakly first-order character accompanied by relatively smaller volume change ΔV than that of the AFI compound of κ -(BEDT-TTF)₂Cu[N(CN)₂]Cl. The evaluated entropy from the Clapeyron equation claims that only the charge degrees of freedom in the Fermi liquid state form gap-like feature as was originally predicted by the Mott's scenario. The spin degrees of freedom may keep the Fermi surface of spinons. These ideas are consistent with the observed thermodynamic data where Fermi-liquid like excitations with gapless (or a tiny gap) characters reported in almost all spin liquids in organics and some intermetallic compounds. This is considered as an important linkage of the spin-liquid physics with the Fermi liquid physics and certainly gives new insight in experimental and theoretical work. However, the following points should be considered to publish in *Nature Communications*.

1. The evaluation of the volume change ΔV of κ -(BEDT-TTF)₂Cu[N(CN)₂]Cl across the phase boundary was determined by the thermal expansion measurement in Ref. 34 (E. Gati *et al.* *Sci. Adv.* (2016)). The authors claim that it is "several ten times smaller" in the case of κ -(BEDT-TTF)₂Cu₂(CN)₃ and performed simulation in Fig.4. Do they have any direct experimental evidence such as dilatometry studies for this which claims important bases to claim that the M-I transition has weakly first order character. This is also important to derive ΔS and discuss $S_{\text{spin liquid phase}}$ in reference to $S_{\text{metallic phase}}$ in the discussion in Fig.4 in the final part of the paper.
2. The critical end point temperature, T_c may be related with the magnitude of electron correlations in the layers. By comparing the J/k_B values in the dimer-Mott lattice of κ -(BEDT-TTF)₂Cu[N(CN)₂]Cl and κ -(BEDT-TTF)₂Cu₂(CN)₃ compounds, there is no drastic difference that indicates the energy scale of the two compounds are almost the same. The authors had better explain why the critical end point in

the frustrated κ -(BEDT-TTF)₂Cu₂(CN)₃ is so much reduced down to the nearly half of that of κ -(BEDT-TTF)₂Cu[N(CN)₂]Cl compound.

3. The inflection point of dT/dp (or dp/dT) of the phase boundary at $T=6$ K and $p=130-135$ MPa in κ -(BEDT-TTF)₂Cu₂(CN)₃ seems to coincide with the point where $T^*(T,p)$ crosses the boundary. The inflection point means that $\Delta S=0$ ($S_{\text{metal phase}}=S_{\text{spin liquid phase}}$) though finite volume change takes place. Is the coincidence accidental or is there any specific reason for the coincidence?
4. In the analysis of ΔS by the Clapeyron formula in Fig.4, the authors assume that ΔV do not have temperature dependence. However, it is natural to consider that ΔV also changes as a function of temperature in both κ -(BEDT-TTF)₂Cu₂(CN)₃ and κ -(BEDT-TTF)₂Cu[N(CN)₂]Cl. If the authors have any information, it is better to describe it.
5. The discussion of Fig.4 also assume that the transition itself is purely an electronic origin in which fractional degrees of electrons forming a fractional gap to change the Fermi liquids to the spin liquids. The phonon contribution to the entropy is not taken into account here. On the other side, recent discussion at ambient pressure for κ -(BEDT-TTF)₂Cu₂(CN)₃ demonstrates that other degrees of freedom which induce dielectric and anomalous phonons are also working to stabilize quantum liquid state. If the structural factor plays a role in spin liquid region, its contribution on the entropy seems to be included in the phase relation at the boundary. Can the authors compare the temperature dependence of spin liquid state of κ -(BEDT-TTF)₂Cu₂(CN)₃ shown in Fig. 4b with the experimental data of the temperature dependence of the spin entropy in the spin liquid phase and discuss how the other degrees of freedom are involved?

In conclusion, the reviewer consider that this paper propose new insight to discuss the gapless spin liquids from experimental side. If the authors can make clear on the above points, I have no hesitation to recommend it publishing in *Nature Communications*.

Reviewer #3 (Remarks to the Author):

In their study "Quasi-continuous transition from a Fermi liquid to a spin liquid", the authors address an important question in the context of quantum spin liquids: What is the nature of the quantum phase transition between a metal and a spin liquid Mott insulator? The field of quantum spin liquids is characterized by a huge mismatch between the number of experimental studies (low) and theoretical ones (high). Any experiment that can yield new understanding or test existing theories is thus well warranted.

The key question, in my opinion, is how much additional insight is gained from the present study compared to earlier work such as Ref. 19 [NatPhys11,221(2015)] written by many of the same authors). The key results advertised in the abstract of the current paper "Here, we demonstrate the case of a quasi-continuous Mott transition from a Fermi liquid to a spin liquid in an organic triangular-lattice system" was already present in Ref. 19, where according to the abstract, the authors "report experimental evidence for the quantum-critical nature of the Mott instability". It appears that in the present study, the authors performed measurements at lower temperatures, which is primarily relevant for studying the properties of the first order transition rather than aspects associated with quantum criticality. Still, it appears to me that the present work does contain new and important information. Firstly, the authors extract new critical exponents, which can be tested against existing or new theoretical studies. Secondly, the authors map out the strength of the weak first order transition, which may be useful for more detailed, quantitative theories.

Given the aforementioned dearth of experimental studies, I therefore still tend towards recommending their work for publication, provided that the authors can make a better case of the novelty and importance of their study. I also append a list of more specific comments below.

Specific comments:

1. In the abstract the authors state that their results "suggest that the spin liquid hosts a spinon Fermi surface, which turns into an electron Fermi surface when charges are Mott delocalized". I believe there are numerous previous experiments that already suggested this and fail to see what the present work does to provide additional support for the existence of a spinon Fermi surface.
2. The critical exponents given in the text (e.g., for the charge gap) do not include error estimates which should be provided.
3. Figure 1 contains typos in the spelling of 'spin liquid'
4. It might be interesting to compare the extracted values of the entropy (for the spin liquid) from the Clausius-Clapeyron analysis to specific heat measurements, which are one of the most compelling pieces of evidence supporting a quantum spin liquid in these compounds.
5. The authors appear to imply that the theory of the metal to spin-liquid Mott transition always predicts a second order transition. E.g., they write "We note, however, that the correspondence between the experiments and the theories is not exact in that $\kappa\text{-(ET)}_2\text{Cu}_2(\text{CN})_3$ exhibits a first-order transition although it is extremely weak." In fact, Ref. 4 discussed the possibility of a first order transition when long-range interactions between electrons are taken into account.
6. The overall level of writing is rather poor and should be improved before the manuscript is suitable for a high-profile journal.

Replies to Reviewer #1's comments

We thank Reviewer 1 for reviewing our manuscript and giving us useful suggestions on how we could improve it. The suggestions have helped us make useful revisions to the manuscript. Below, we provide replies to each point and describe the revised parts in the manuscript, which are highlighted by red characters in the revised manuscript.

[Comment 1-1]

One can read at l. 126: "...the transport characteristics around the Mott boundary are featured by a substantial falloff in T^* , the critical behaviors of the coefficient A , the closing of charge gap Δ , and the resistance plateau, all of which go against the conventional first-order Mott transition but accord with the theoretical consequences of the Mott transition of a spin liquid with a spinon Fermi surface."

The decrease of T^* and the divergent behavior of the coefficient A are typical signatures for conventional first-order Mott transition both related to the divergency of the electronic effective mass as predicted theoretically (Rev. Mod. Phys. 68, 13–125 (1996) and references therein) and demonstrated in several experiments.

As a consequence, only "the closing of charge gap Δ and the resistance plateau go against the conventional first-order Mott transition but accord with the theoretical consequences of the Mott transition of a spin liquid with a spinon Fermi surface".

The related text in the article needs to be changed accordingly.

[Reply 1-1]

As the reviewer pointed out, the coefficient A increases in a seemingly critical manner and T^* decreases on approaching the Mott metal-insulator transition even in conventional systems. However, in the vicinity of the first-order transition line of the conventional Mott transition, A and T^* should not behave in a critical manner owing to the discontinuous nature of the transition. In fact, the coefficient A appears to show a divergent increase in a crude pressure scale for κ -(ET)₂[N(CN)₂]Cl, κ -(ET)₂[N(CN)₂]Br, and deuterated κ -(ET)₂[N(CN)₂]Br, all of which exhibit the conventional 1st-order Mott transition; however, the diverging behavior of A becomes weak in the vicinity of the Mott transition, as shown in Fig. S8 in the revised Supplementary Information. In the present system, the coefficient A continues to increase in a critical manner even at $P-P_c \lesssim 50$ MPa, indicating the unconventional quasi-continuous nature of the Mott transition between a spin liquid and a Fermi liquid. We note that T^* at the Mott boundary is as low as ~ 5 K in the present system while it is ~ 30 K in κ -(ET)₂[N(CN)₂]Cl (Ref.

S7). However, thanks to this comment, we noticed the lack of the above discussion in the original manuscript, and so we have revised the manuscript, as described below.

In the 5th paragraph in the revised manuscript, we have added the text,

“The divergent behaviour of A persists in the very vicinity of P_c , $|P-P_c| < 50$ MPa, whereas the increase in A becomes weak in the vicinity of the conventional Mott transition (see Fig. S8).”

In the revised Supplementary Information, we have added Fig. S8 with additional references,

- S7. Limelette, P. *et al.* Mott Transition and Transport Crossovers in the Organic Compound κ -(BEDT-TTF)₂Cu[N(CN)₂]Cl. *Phys. Rev. Lett.* **91**, 016401 (2003).
- S8. Sasaki, T., Yoneyama, N. & Kobayashi, N. Mott transition and superconductivity in the strongly correlated organic superconductor κ -(BEDT-TTF)₂Cu[N(CN)₂]Br. *Phys. Rev. B* **77**, 054505 (2008).
- S9. Strack, C. *et al.* Resistivity studies under hydrostatic pressure on a low-resistance variant of the quasi-two-dimensional organic superconductor κ -(BEDT-TTF)₂Cu[N(CN)₂]Br: Search for intrinsic scattering contributions. *Phys. Rev. B* **72**, 054511 (2005).

[Comment 1-2]

(A) It results then that the closing of charge gap Δ and the resistance plateau are two crucial features calling for improved analysis and discussions in order to demonstrate a "Quasi-continuous transition from a Fermi liquid to a spin liquid".

Concerning the closing of charge gap, Fig. 2c displays an Arrhenius plot of resistance in the insulating phase with $\ln(R)$ as a function of $1/T$. One cannot judge the validity of the performed fits especially in the regions of interest, namely where the charge gap is small. A plot of the derivative of resistance as $2*d(\ln(R))/d(1/T)$ as a function of temperature (namely the gap as a function of T), or equivalent representation, could better demonstrate the vanishing of the gap.

(B) In addition, Fig. 3 shows the pressure dependence of the gap with very small values in the regions around 120 MPa, including likely $\Delta/2 < T$. In such a regime, resistance is usually not activated because thermal excitations are higher than the charge gap. Thus, the authors have to explain the method used to extract the gap within this regime of small values in order to

demonstrate its vanishing.

[Reply 1-2]

(A) Following the reviewer's suggestion, we performed a further analysis to evaluate the charge gap. We plotted the $2 \times d(\ln R)/d(1/T)$ as a function of $1/T$ (Fig. S9 in the revised Supplementary Information). The charge gap is defined by the dashed lines shown in Fig. S9. The newly evaluated values of the charge gap approximately reproduced the original values. We replaced the original values by the new values in the revised manuscript and revised the manuscript as follows.

In the 6th paragraph, we have replaced the original text,

"...by a charge gap, Δ , obtained by fitting the form of $R(T) = C \exp(\Delta/2T)$ to the data (Fig. 2c). A remarkable feature..."

by the new one,

"...by a charge gap, Δ , that is deduced from the value of $2d(\ln R)/d(1/T)$ at low temperatures (see the Supplementary Information for details). As shown in Fig. 2c and Fig. 3, a remarkable feature...",

and we added the following new subsection in the revised Supplementary Information.

"Estimation of charge gap

Because the charge gap Δ can be extracted from the relation $R \propto \exp(\Delta/2T)$ at $T \ll 2\Delta$, $2d(\ln R)/d(1/T)$ directly yields the value of the charge gap. We plot the $2d(\ln R)/d(1/T)$ as a function of $1/T$ in Fig. S9. The charge gap is defined by the dashed lines shown in Fig. S9. At $P > 118$ MPa, it is difficult to extract the value of charge gap from the $2d(\ln R)/d(1/T)$ because $2d(\ln R)/d(1/T)$ is much less than $2T$. However, it is obvious that the temperature dependence of resistance at $P > 118$ MPa showed vanishingly small gaps that were much smaller than the lowest measured temperature of ~ 2 K."

(B) We agree with the reviewer that the charge gap should not be extracted from the resistance at temperatures higher than $\Delta/2$ and thus the estimated gaps at $P > 118$ MPa in the original manuscript are ambiguous. However, it is obvious that the temperature dependence of resistance at $P > 118$ MPa shows vanishingly small gaps much less than the lowest temperature measured of ~ 2 K; so, we think that it is not too much to say that the charge gap is almost closed at $P >$

118 MPa (Figs.2a, 2c). Accordingly, we have removed the data points of the charge gap at $P > 118$ MPa from Figs. 1 and 3; instead, we have highlighted the pressure region where the charge gap is almost closed in the revised Fig. 3.

[Comment 1-3]

Concerning the resistance plateau, the reported value is higher than the predicted one as stated by the Authors but this resistance behavior appears qualitatively distinct from the resistance behavior across a conventional Mott transition. Then, it could be consistent with a Mott transition from a Fermi liquid to a spin liquid at a qualitative level. Nevertheless, a non monotonic resistance behavior is clearly seen in Fig. 2a both below and above 120 MPa (with a relative maximum of $R(T)$) which is neither really discussed by the Authors and nor theoretically expected.

In particular, there is a sizable decrease of resistance in the Mott insulating state at 118 MPa for instance as a function of decreasing temperature which is very unusual and requires at least some comments from the Authors.

[Reply 1-3]

The non-monotonic temperature dependence of resistance near 120 MPa, which appears strange, is reasonable, considering the bending of the quasi-continuous Mott-transition line towards higher pressures with increasing temperature. As the temperature decreases from approximately 15 K to several kelvins at a fixed pressure of $P \lesssim 135$ MPa, the system approaches the metallic phase, because the bent first-order transition line ($dT/dP > 0$) nears the fixed pressure at low temperatures (see Fig. 1a). In addition, the present Mott transition has a quasi-continuous nature. Thus, the approach to the Mott boundary by cooling causes a decrease in the charge gap in the insulating side and possibly results in a decrease in resistance with temperature. Eventually, the resistance increases on further cooling because the system is located in the Mott insulating side at a finite distance from the Mott transition line, which is nearly vertical at temperatures below ~ 5 K. We have added the above discussion into the 4th paragraph of the revised manuscript.

[Comment 1-4]

Furthermore, the Authors reproduce in Fig. 1b and 1c a Quantum critical region over a wide temperature range. The Authors should precise how this high temperature Quantum critical region is connected to the low temperature Quantum critical region as theoretically discussed in Phys. Rev. B 86, 321 245102 (2012), the latter being defined with a temperature independent

resistance not supported by experimental results over such a wide temperature range.

The suggested role of disorder at l. 122 to explain all the discrepancies with theoretical predictions needs to be clarified in order to provide a consistent description of the phase diagrams.

[Reply 1-4]

We have also been very much concerned with the relation between the high-temperature and low-temperature quantum criticalities, but remained having no convincing idea. Stimulated by the reviewer's comment, however, we reconsidered this issue and reached a thought although not definitive. In the revised manuscript, we mentioned our thought for stimulating future research on this issue by adding the following sentences as the last paragraph, "*Finally, we discuss the relation between the low-temperature (Low-T) quantum criticality observed here and the high-temperature (high-T) quantum criticality (Figs. 1b, c). The latter has the following key features. First, it is commonly observed for various systems irrespective of lattice geometry, whether the Mott insulator is a spin liquid or an antiferromagnet or whether the Mott transition is of the first order or quasi-continuous. Second, the high-T regime should have no memories about Fermi surfaces in the metallic side and the nature of the magnetic ground state in the Mott insulating side. Third, the high-T quantum critical scaling of resistivity is reproduced by the single-site DMFT of the Hubbard model. All the above suggest that the high-T quantum criticality is a universal high-energy characteristic of the Mott transition. In contrast, the low-T quantum criticality is likely specific to the fluctuations between electron Fermi surfaces and spinon Fermi surfaces, being closely related to the kinds of ground states. In our tentative view, the Mott transition has a hierarchal energy structure. The high-energy quantum criticality is generic, whereas the low-energy one depends on the details of the lattice, and the former is characterized by the scaling function fulfilled by resistivity in contrast to the constant resistivity in the latter. The proximity of the $z\nu$ values of the two criticalities may provide some insight into their possible connection, which awaits theoretical exploration.*"

In general, the effect of disorder becomes significant especially at low temperatures. As we wrote in the original manuscript, we consider that the possible role of disorder in the present case is extending the critical point into a finite region.

[Comment 1-5]

Finally, the Authors propose an analysis of thermodynamic properties of spin liquid on the Mott boundary by extracting entropy, based on several assumptions and considering the volume

change as a constant free parameter.

Also, one can read at l. 138: "The $dP_{\text{jump}}/dT_{\text{jump}}$ determined from the fitting curve of the phase boundary (Fig. 1a) gives the temperature dependence of $S_{\text{QSL,spin}} - S_{\text{metal,el}}$ with ΔV as a parameter (Fig. 4a). Because ΔV and P are thermodynamically conjugate variables, ΔV plays a role of order parameter of Mott transition and therefore measures the strength of the first-order transition."

If the volume change "plays a role of order parameter of Mott transition" as stated, it must be temperature dependent in contrast to the assumption made by the Authors which has allowed them to infer the temperature dependence of entropy in Fig. 4. Therefore, the Authors have to reconcile their point of view with their analysis or at least precise that the analysis of the deduced entropy must only be considered at a qualitative level despite its interest.

[Reply 1-5]

In the present Clausius-Clapeyron analysis, the temperature ranges required for deducing ΔV through considering the entropy values are well below T_c ($T < 6 \text{ K} \sim 0.3T_c$ for $\kappa\text{-(ET)}_2\text{Cu}_2\text{(CN)}_3$ and $T < 20 \text{ K} \sim 0.5T_c$ for $\kappa\text{-(ET)}_2\text{Cu[N(CN)}_2\text{]Cl}$). For $0.3 T_c - 0.5 T_c$ or lower temperatures, the order parameter, namely ΔV , is expected to be practically constant, which is why we did not take its temperature dependence into consideration. At higher temperatures, ΔV should be temperature-dependent more or less. To secure the entropy analysis even at high temperatures near the critical endpoint (although not affecting the value of ΔV), we performed the Clausius-Clapeyron analysis with incorporating the temperature dependence of ΔV , following the reviewer's suggestion. We described the details of this reanalysis in revised Supplementary Information. As the two-dimensional Mott transition can be mapped to the two-dimensional Ising model, we adopted the temperature profile of the order parameter of the two-dimensional Ising model as the temperature dependence of ΔV , which is shown in Figs. S6a and S6d. It is seen that the ΔV varies only at highest temperatures measured, which makes no practical differences between Figs. 4a-d (assuming $\Delta V = \text{const.}$) and Figs. S6 b,c,e,f (assuming the above-mentioned T -dependence of ΔV), respectively.

We added the following text in the 7th paragraph of the revised manuscript,

"We also performed a Clausius-Clapeyron analysis while incorporating the temperature dependence of ΔV and found the results in Fig. 4 to be practically unaltered (see the Supplementary Information for details).",

and added the following new subsection in the revised Supplementary Information with additional supplementary Fig. S6.

“The Clausius-Clapeyron analysis taking into account the temperature dependence of the volume jump ΔV

In the main text, we assume that ΔV is temperature independent well below T_c because ΔV is reasonably expected to saturate at low temperatures. Here, we conduct a Clausius-Clapeyron analysis for $\kappa\text{-(ET)}_2\text{Cu}_2(\text{CN})_3$ and $\kappa\text{-(ET)}_2\text{Cu}[\text{N}(\text{CN})_2]\text{Cl}$, taking into account the temperature dependence of ΔV . There are no direct experimental results of $\Delta V(T)$. We then adopted the temperature profile of the order parameter of the two-dimensional Ising model as the temperature dependence of ΔV because the two-dimensional Mott transition can be mapped to the two-dimensional Ising model. Figures S6a and S6d show the expected temperature dependencies of ΔV for $\kappa\text{-(ET)}_2\text{Cu}_2(\text{CN})_3$ and $\kappa\text{-(ET)}_2\text{Cu}[\text{N}(\text{CN})_2]\text{Cl}$, respectively. It is seen that the ΔV varied only at the highest measured temperatures, resulting in no practical differences between Figs. 4a-d (assuming $\Delta V=\text{const.}$) and Figs. S6 b, c, e, and f (assuming the abovementioned T -dependence of ΔV), respectively.”

Replies to Reviewer #2's comments

We thank Reviewer 2 for reviewing our manuscript and providing illuminating comments. The constructive suggestions helped us to improve our manuscript. Below, we reply point-by-point to the comments and describe the revised parts in the manuscript, which are highlighted by red characters in the revised manuscript.

[Comment 2-1]

The evaluation of the volume change ΔV of k -(BEDT-TTF)₂Cu[N(CN)₂]Cl across the phase boundary was determined by the thermal expansion measurement in Ref. 34 (E. Gati et al. *Sci. Adv.* (2016)). The authors claim that it is “several ten times smaller” in the case of k -(BEDT-TTF)₂Cu₂(CN)₃ and performed simulation in Fig.4. Do they have any direct experimental evidence such as dilatometry studies for this which claims important bases to claim that the M-I transition has weakly first order character. This is also important to derive ΔS and discuss S_{spin} liquid phase in reference to S_{metallic} phase in the discussion in Fig.4 in the final part of the paper.

[Reply 2-1]

As the reviewer pointed out, it is desirable that ΔV of κ -(ET)₂Cu₂(CN)₃ is determined directly by experiments. However, there are no such experiments to date. This is because a dilatometer works only in the liquid phase of helium (pressure-transmitting) medium. Unfortunately, the first order transition line of κ -(ET)₂Cu₂(CN)₃ is located in the solid phase of the pressure-temperature phase diagram of helium. Indeed, we had considered the same thing as the reviewer suggested and consulted one of the authors of the paper of E. Gati et al. *Sci. Adv.* (2016) about the feasibility of the dilatometry experiment for κ -(ET)₂Cu₂(CN)₃. His reply was that the experiment would be impossible by the above-mentioned reason.

[Comment 2-2]

The critical end point temperature, T_c may be related with the magnitude of electron correlations in the layers. By comparing the J/k_B values in the dimer-Mott lattice of k -(BEDT-TTF)₂Cu[N(CN)₂]Cl and k -(BEDT-TTF)₂Cu₂(CN)₃ compounds, there is no drastic difference that indicate the energy scale of the two compounds are almost the same. The authors had better explain why the critical end point in the frustrated k -(BEDT-TTF)₂Cu₂(CN)₃ is so much reduced down to the nearly half of that of k -(BEDT-TTF)₂Cu[N(CN)₂]Cl compound.

[Reply 2-2]

Since the critical endpoint of the Mott transition in quasi-two-dimensional organic conductors was first identified by a French group in 2000, we have been interested in the suggested problem, “what factors determine the critical endpoint”. One of the authors (K. Kanoda) had had several opportunities to discuss this question with theorists, but the problem was still open. However, as we mentioned in the manuscript, a recent numerical study (Ref. 2) has indicated that, in the Mott transition between a Fermi liquid and a spin liquid, the critical endpoint is pushed down to absolute zero, resulting in a quantum phase transition. This means that the degree of frustration is undoubtedly one of the prime factors determining the critical temperature of Mott transition. It is hoped to see the variation of the critical endpoint under a systematic control of the frustration parameter. To the best of our knowledge, there are no theoretical and experimental studies answering this issue. (I remember that a DMFT researcher mentioned the endpoint was insensitive to anisotropy of the transfer integrals as far as the Cluster-DMFT is concerned.) We hope future progress on this issue from both of theories and experiments.

[Comment 2-3]

The inflection point of dT/dp (or dp/dT) of the phase boundary at $T=6$ K and $p=130-135$ MPa in $k\text{-(BEDT-TTF)}_2\text{Cu}_2(\text{CN})_3$ seems to coincide with the point where $T^*(T,p)$ crosses the boundary. The inflection point means that $\Delta S=0$ ($S_{\text{metal phase}}=S_{\text{spin liquid phase}}$) though finite volume change takes place. Is the coincidence accidental or is there any specific reason for the coincidence?

[Reply 2-3]

We appreciate the reviewer’s incisive comment. A similar anomaly suggesting the rapid release of the spin entropy is observed in the specific heat around 6 K at ambient pressure (Ref.10). One possible origin of the entropy reduction discussed there is the formation of a spinon Fermi surface from an incoherent paramagnetic state around 6 K. The present study suggests that the similar thing may happen at the similar temperature (at the Mott boundary) under pressure. If this scenario is the case, it turns out that the electron Fermi surface in the metallic phase and the spinon Fermi surface in the spin liquid phase take shape at similar temperatures. It is an interesting argument, so we mentioned it in the 10th paragraph in the revised manuscript in a cautious manner since the origin of the 6 K anomaly at ambient pressure is under debate;
“It is suggestive that the temperature of the inflection point in the entropy decrease, 6 K, roughly coincides with T^ in the metallic side at the Mott boundary. A similar anomaly suggesting the*

rapid release of the spin entropy was observed in the specific heat at approximately 6 K at ambient pressure (Ref. 10), and the formation of a spinon Fermi surface from an incoherent paramagnetic state was argued as a possible origin. The present results may suggest that the electron Fermi surface in the metallic phase and the spinon Fermi surface in the spin liquid phase take shape at similar temperatures, although the origin of the 6 K anomaly at ambient pressure is under debate.”

[Comment 2-4]

In the analysis of ΔS by the Clapeyron formula in Fig.4, the authors assume that ΔV do not have temperature dependence. However, it is natural to consider that ΔV also changes as a function of temperature in both κ -(BEDT-TTF)₂Cu₂(CN)₃ and κ -(BEDT-TTF)₂Cu[N(CN)₂]Cl. If the authors have any information, it is better to describe it.

[Reply 2-4]

We agree with the reviewer’s suggestion. We had the same suggestion from Reviewer #1, too; so we duplicate our reply to Reviewer#1 (Reply 1-5) below.

In the present Clausius-Clapeyron analysis, the temperature ranges required for deducing ΔV through considering the entropy values are well below T_c ($T < 6 \text{ K} \sim 0.3T_c$ for κ -(ET)₂Cu₂(CN)₃ and $T < 20 \text{ K} \sim 0.5T_c$ for κ -(ET)₂Cu[N(CN)₂]Cl). For $0.3 T_c - 0.5 T_c$ or lower temperatures, the order parameter, namely ΔV , is expected to be practically constant, which is why we did not take its temperature dependence into consideration. At higher temperatures, ΔV should be temperature-dependent more or less. To secure the entropy analysis even at high temperatures near the critical endpoint (although not affecting the value of ΔV), we performed the Clausius-Clapeyron analysis with incorporating the temperature dependence of ΔV , following the reviewer’s suggestion. We described the details of this reanalysis in revised Supplementary Information. As the two-dimensional Mott transition can be mapped to the two-dimensional Ising model, we adopted the temperature profile of the order parameter of the two-dimensional Ising model as the temperature dependence of ΔV , which is shown in Figs. S6a and S6d. It is seen that the ΔV varies only at highest temperatures measured, which makes no practical differences between Figs. 4a-d (assuming $\Delta V = \text{const.}$) and Figs. S6 b,c,e,f (assuming the above-mentioned T -dependence of ΔV), respectively.

We added the following text in the 7th paragraph of the revised manuscript,

“We also performed a Clausius-Clapeyron analysis while incorporating the temperature dependence of ΔV and found the results in Fig. 4 to be practically unaltered (see the Supplementary Information for details).”

and added the following new subsection in the revised Supplementary Information with additional supplementary Fig. S6.

“The Clausius-Clapeyron analysis taking into account the temperature dependence of the volume jump ΔV

In the main text, we assume that ΔV is temperature independent well below T_c because ΔV is reasonably expected to saturate at low temperatures. Here, we conduct a Clausius-Clapeyron analysis for $\kappa\text{-(ET)}_2\text{Cu}_2(\text{CN})_3$ and $\kappa\text{-(ET)}_2\text{Cu}[\text{N}(\text{CN})_2]\text{Cl}$, taking into account the temperature dependence of ΔV . There are no direct experimental results of $\Delta V(T)$. We then adopted the temperature profile of the order parameter of the two-dimensional Ising model as the temperature dependence of ΔV because the two-dimensional Mott transition can be mapped to the two-dimensional Ising model. Figures S6a and S6d show the expected temperature dependencies of ΔV for $\kappa\text{-(ET)}_2\text{Cu}_2(\text{CN})_3$ and $\kappa\text{-(ET)}_2\text{Cu}[\text{N}(\text{CN})_2]\text{Cl}$, respectively. It is seen that the ΔV varied only at the highest measured temperatures, resulting in no practical differences between Figs. 4a-d (assuming $\Delta V=\text{const.}$) and Figs. S6 b, c, e, and f (assuming the abovementioned T -dependence of ΔV), respectively.”

[Comment 2-5]

The discussion of Fig.4 also assume that the transition itself is purely an electronic origin in which fractional degrees of electrons forming a fractional gap to change the Fermi liquids to the spin liquids. The phonon contribution to the entropy is not taken into account here. On the other side, recent discussion at ambient pressure for $\kappa\text{-(BEDT-TTF)}_2\text{Cu}_2(\text{CN})_3$ demonstrates that other degrees of freedom which induce dielectric and anomalous phonons are also working to stabilize quantum liquid state. If the structural factor plays a role in spin liquid region, its contribution on the entropy seems to be included in the phase relation at the boundary. Can the authors compare the temperature dependence of spin liquid state of $\kappa\text{-(BEDT-TTF)}_2\text{Cu}_2(\text{CN})_3$ shown in Fig. 4b with the experimental data of the temperature dependence of the spin entropy in the spin liquid phase and discuss how the other degrees of freedom are involved?

[Reply 2-5]

The reviewer's comment is very inspiring for further looking into the nature of the spin liquid. The suggested method will work if we can estimate the spin entropy in the spin liquid phase correctly. In conventional paramagnetic cases, spin entropy can be derived from the spin susceptibility. Although the spin susceptibility of the spin liquid phase of κ -(ET)₂Cu₂(CN)₃ is available, there are several difficulties in estimating the spin entropy from it. First, the spin liquid state of the present material shows anomalous field-dependent magnetism. The studies of NMR and μ SR revealed that applying external magnetic field induces a small and inhomogeneous staggered moment, even when a field is much less than 1 T, and the inhomogeneous moment is proportional to an external magnetic field, clearly distinguished from the ordinary magnetic ordering. Second, in relation to the above, it is generally accepted that the spin liquid, which is a non-ordered state in spite of possessing strong spin correlation, is in an anomalous spin state distinct from the conventional paramagnetic state; the presence of chiral order is an example of theoretical proposals. Taking these aspects of the present system into consideration, we think it is not reasoned at all to extract the spin entropy from the spin susceptibility observed under a finite magnetic field of the order of 1 Tesla by the method applied to ordinary paramagnetic states. To avoid misleading arguments, we refrain from going to the above analysis at present.

Replies to Reviewer #3's comments

We would like to thank Reviewer 3 for his/her careful reading of our manuscript and his/her useful suggestions. His/her deep insight into Mott transition and a quantum spin liquid helped us improve our manuscript. Below, we reply point-by-point to the comments and describe the revised parts in the manuscript, which are highlighted by red characters in the revised manuscript.

[Comment 3-1]

In the abstract the authors state that their results “suggest that the spin liquid hosts a spinon Fermi surface, which turns into an electron Fermi surface when charges are Mott delocalized”. I believe there are numerous previous experiments that already suggested this and fail to see what the present work does to provide additional support for the existence of a spinon Fermi surface.

[Reply 3-1]

As the reviewer pointed out, there are several experimental studies suggesting the existence of a spinon Fermi surface. However, we have to say that not a few people think it is still under debate and it is a consensus that there are no smoking-gun experiments on this issue to date. In this situation, it is meaningful to tackle this issue from different viewpoints. The present study is the first to provide discussion on this problem through examining the nature of the Mott transition. Having the referee's comment, we revised the abstract to make clear the suggested point as follows:

We have inserted a sentence,

“Indeed, several experimental studies have suggested the existence of such spin liquids⁵.”

before the sentence starting from “Here” in the abstract.

We modified the last sentence in the abstract as follows.

Before revision: *“These results suggest that the spin liquid hosts a spinon Fermi surface, which turns into an electron Fermi surface when charges are Mott delocalized.”*

After revision: *“These results provide additional support for the existence of a spinon Fermi surface, which becomes an electron Fermi surface when charges are Mott delocalized.”*

[Comment 3-2]

The critical exponents given in the text (e.g., for the charge gap) do not include error estimates which should be provided.

[Reply 3-2]

According to this suggestion, we have estimated the errors of the critical exponents of A and Δ : $A \propto |P-P_c|^{-0.75 \pm 0.01}$ and $\Delta \propto |P-P_c|^{0.73 \pm 0.04}$, which are shown in the revised manuscript. Note that we have newly evaluated the charge gap by performing the analysis suggested in Comment [1-2]. We have also added the error bars to the data points of A and Δ in Fig. 1c and Fig. 3.

[Comment 3-3]

Figure 1 contains typos in the spelling of ‘spin liquid’

[Reply 3-3]

We have fixed the typos.

[Comment 3-4]

It might be interesting to compare the extracted values of the entropy (for the spin liquid) from the Clausius-Clapeyron analysis to specific heat measurements, which are one of the most compelling pieces of evidence supporting a quantum spin liquid in these compounds.

[Reply 3-4]

We interpret the reviewer’s comment as suggesting us to extract the specific heat from the entropy deduced from the Clausius-Clapeyron analysis and to compare it with the specific heat directly measured at ambient pressure. In fact, we had performed that analysis. To obtain specific heat through the Clausius-Clapeyron relation, we need the temperature derivative of the entropy, that is, the second derivative of the Mott transition line, which is quite sensitive to the fitting of the experimental Mott transition line while the entropy, which is given by the first derivative of the transition curve, is reliable. Thus, we were afraid that the evaluated specific heat had large ambiguity. Receiving this comment, however, we decided to show the derived specific heat in the revised Supplementary Information for reference, because the deduced specific heat divided by temperature (Fig. S7) shows a peak around 6-7 K akin to the ambient-pressure data, suggesting that the 6-K anomaly in the ambient pressure persists up to the critical

pressure of the Mott transition. We think that this qualitative feature of the peak formation is not affected by the curve fitting of the Mott boundary.

We displayed the specific heat values deduced by the Clausius-Clapeyron analysis in Fig. S7 with a caption explaining the way of analysis in the revised Supplementary Information. Accordingly, we inserted the following text in the 8th paragraph of the revised manuscript,

“We also estimated the specific heat from $S_{QSL,spin}$, as shown in Fig. S7 in the Supplementary Information. Although the specific heat is less reliable since it is given by the second derivative of the fitting curve of the phase boundary, the deduced specific heat appears to show a qualitative feature, that is, peak formations at approximately 6-7 K, akin to the ambient-pressure behaviour (Ref.10) and suggesting that the 6-K anomaly at ambient pressure persists up to the critical pressure of the Mott transition.”

[Comment 3-5]

The authors appear to imply that the theory of the metal to spin-liquid Mott transition always predicts a second order transition. E.g., they write “We note, however, that the correspondence between the experiments and the theories is not exact in that κ -(ET)₂Cu₂(CN)₃ exhibits a first-order transition although it is extremely weak.” In fact, Ref. 4 discussed the possibility of a first order transition when long-range interactions between electrons are taken into account.

[Reply 3-5]

We missed the point notified by the reviewer. To incorporate the suggestion in the manuscript, we have revised the sentences in the 11th paragraph as follows.

Before revision: *“We note, however, that the correspondence between the experiments and the theories is not exact in that κ -(ET)₂Cu₂(CN)₃ exhibits a first-order transition although it is extremely weak. This may indicate that some additional factors not included in the theoretical treatments affect the ground states; indeed, superconductivity appears in the metallic phase and the 6-K anomaly possibly suggestive of some instability of the spin liquid emerges in the insulating phase. Very recently, a DMFT-based theoretical treatment incorporating spinon excitations suggest the first-order nature of the Mott transition of a spinon Fermi surface. The faint remanence of the first -order nature may have some relevance to this theoretical consequence³⁵. In any event, the present observations distinct from the conventional cases are expected to promote further theoretical studies.”*

After revision: “We note, however, that the correspondence between the experiments and theories based on the Hubbard model is not exact in that $\kappa\text{-(ET)}_2\text{Cu}_2(\text{CN})_3$ exhibits a first-order transition, although it is extremely weak. This may indicate that some additional factors not included in the theoretical treatments affect the ground states; indeed, superconductivity appears in the metallic phase, and the 6-K anomaly is possibly suggestive of some instability of the spin liquid emerging in the insulating phase. In addition, the possible first-order transition due to long-range interactions among electrons is theoretically discussed in Ref. 4. Very recently, a DMFT-based theoretical treatment incorporating spinon excitations suggest the first-order nature of the Mott transition of a spinon Fermi surface³⁵. The faint remanence of the first-order nature may have some relevance to these theoretical consequences^{4,35}.”

[Comment 3-6]

The overall level of writing is rather poor and should be improved before the manuscript is suitable for a high-profile journal.

[Reply 3-6]

Having the above comments, we consulted native speakers for improving the original manuscript.

REVIEWERS' COMMENTS:

Reviewer #1 (Remarks to the Author):

The Authors have suitably addressed my main comments so I recommend to publish their paper.

Reviewer #2 (Remarks to the Author):

I read the revised manuscript and supplementary information of the article by T. Furukawa and K. Kanoda et al. on the transport and thermodynamic discussion on the spin liquid compound of κ -(BEDT-TTF)₂Cu₂(CN)₃. I also checked the replies to the questions I have raised in the previous manuscript. The response to each point and revisions of manuscript marked by red color sound reasonable. I agree that the point 2-5 is important for future discussion but quite difficult to do experiments under the gas-pressure condition. However, the constructive comment the authors included in p.13 in the revised manuscript (the discussion on high-T and low-T regime) suggests something related to this point. I think it is worth publishing in Nature Communications.